

# Impact of COVID-19 lockdown on air quality analyzed through machine learning techniques

Umer Zukaib[1,2], Mohammed Maray[3], Saad Mustafa[1], Nuhman Ul Haq[1], Atta ur Rehman Khan[4] and Faisal Rehman[1]

[1] Computer Science, COMSATS University Islamabad, Abbottabad Campus, Abbottabad, KP, Pakistan
[2] Key Laboratory of Aerospace Information Security and Trusted Computing, Ministry of Education, School of Cyber Science and Engineering, Wuhan University, Wuhan, China
[3] College of Computer Science and Information Systems, King Khalid University, Abha, Saudi Arabia
[4] College of Engineering and Information Technology, Ajman University, Ajman, UAE

Corresponding author
Saad Mustafa, saad-mustafa@cuiatd.edu.pk

## ABSTRACT

After February 2020, the majority of the world's governments decided to implement a lockdown in order to limit the spread of the deadly COVID-19 virus. This restriction improved air quality by reducing emissions of particular atmospheric pollutants from industrial and vehicular traffic. In this study, we look at how the COVID-19 shutdown influenced the air quality in Lahore, Pakistan. HAC Agri Limited, Dawn Food Head Office, Phase 8-DHA, and Zeenat Block in Lahore were chosen to give historical data on the concentrations of many pollutants, including $PM2.5$, $PM10$ (particulate matter), $NO_2$ (nitrogen dioxide), and $O_3$ (ozone). We use a variety of models, including decision tree, SVR, random forest, ARIMA, CNN, N-BEATS, and LSTM, to compare and forecast air quality. Using machine learning methods, we looked at how each pollutant's levels changed during the lockdown. It has been shown that LSTM estimates the amounts of each pollutant during the lockout more precisely than other models. The results show that during the lockdown, the concentration of atmospheric pollutants decreased, and the air quality index improved by around 20%. The results also show a 42% drop in $PM2.5$ concentration, a 72% drop in $PM10$ concentration, a 29% drop in $NO_2$ concentration, and an increase of 20% in $O_3$ concentration. The machine learning models are assessed using the RMSE, MAE, and R-SQUARE values. The LSTM measures $NO_2$ at 4.35%, $O_3$ at 8.2%, $PM2.5$ at 4.46%, and $PM10$ at 8.58% in terms of MAE. It is observed that the LSTM model outperformed with the fewest errors when the projected values are compared with the actual values.

## INTRODUCTION

The COVID-19 pandemic has produced major problems in health and socioeconomic activity all throughout the world (*Cobb et al., 2021*; *Aljohani et al., 2021*; *Hassan et al., 2022*). Certain cofactors, such as hazardous atmospheric pollutants, should be considered in the context of disease spread and increasing fatality rates, particularly in densely

populated regions (*Conticini, Frediani & Caro, 2020*; *Bello-Chavolla et al., 2021*). According to one study, the infection rates of the fatal virus COVID-19 and atmospheric pollutants including PM10, CO, SO2, and NH3 have a direct association; as pollution levels rise, so do COVID-19 infection rates (*Zhu et al., 2020*; *Srivastava, 2021*). Lockdown, on the other hand, dramatically improves air quality (*Bhatti et al., 2022*). According to a Chinese study (*Zhang et al., 2022*), the concentration of air pollutants reduced by 5 to 24 percent during the lockdown, with the decline reaching 40 percent in megacities. Another study of megacities found that lockout reduces PM2.5 concentrations by 20 to 60 percent (*Rodríguez-Urrego & Rodríguez-Urrego, 2020*). Furthermore, another environmental study discovered that the fall in industrial activity and transportation is to blame for the rapid drop in air pollution levels (*Kerimray et al., 2020*; *Sbai et al., 2021*).

Lockdown has been criticized for various economic crises and is regarded to be a barrier to progress. It decreases the transmission of viruses and reduces pollutants, but it has a long-term negative impact on economic growth (*Mahmud & Riley, 2021*). In the majority of countries, local authorities imposed the lockdown. Therefore, the research community will now have a once-in-a-lifetime chance to investigate the impacts of lockdown on air quality (*Desvars-Larrive et al., 2020*). Utilizing trajectory-based models, some research investigated the impact of lockdown on the quality of the air (*Šimić et al., 2020*; *Zhao et al., 2020*). Researchers have also looked at the independent factors that affect air quality using fixed-effect models (*Liu et al., 2020*; *Venter et al., 2020*). The majority of nations have seen an improvement in air quality during these lockdown times. The concentrations of several pollutants, including PM2.5, NO2, SO2, and CO2, are noticeably lowered in Wuhan, China, during the lockdown (*Al-qaness et al., 2021*). Various studies have presented air quality prediction and forecasting models both during and after lockdown. However, bidirectional LSTM and encoder–decoder based LSTM with multivariate data showed better results (*Mao et al., 2021*).

In this study, we estimate air quality using decision tree regression, support vector regression, random forest regression, auto-regressive integrated moving average (ARIMA), convolution neural networks (CNN), N-BEATS, and long short-term memory (LSTM). Machine learning approaches are used to investigate the improvement in air quality during the COVID-19 shutdown in Lahore, Pakistan. Although all of the models yielded excellent results, the LSTM model performed the best due to its superior multivariate problem-solving capabilities. We examined historical air quality data from four locations in Lahore, Pakistan, namely "HAC Agri Limited", "Dawn Food Head Office", "Phase 8-DHA", and "Zeenat Block", which included average concentrations of several pollutants: PM2.5, PM10 (particulate matter), NO2 (nitrogen dioxide), and O3 (ozone). For the LSTM model to forecast these contaminants' anticipated concentrations during the lockdown period, the average concentration of these pollutants is set as a goal variable. The LSTM model may be generalized well. As a result, LSTM is an appropriate deep learning model for analyzing the decrease in pollutant concentration during the COVID-19 lockout. The LSTM model is used to analyze historical air quality data. LSTM used training data from January 2018 through December 2019. A separate dataset, the validation set, is used for validation. The data is covered from January 2020 to May 2020, although

the actual lockdown time is from March 2020 to May 2020. The model also projected the four-day moving average for many air pollutants. The results reveal that the LSTM model provides more accurate predictions with fewer errors.

Major contributions of this article are:

- To determine the main traits, we used principal component analysis (PCA) on the AQI dataset.
- We used scatter plots and the Mahalanobis Distance (MD) technique to find outliers and impute their values using the column mean.
- Several machine learning algorithms were used to forecast the concentration of various pollutants.

The rest of the article is organized as follows: Section 2 presents the related work. Section 3 shows the material and methods. Section 4 presents the experimentation and results. Section 5 presents conclusions and future work.

## RELATED WORK

Atmospheric pollutants such carbon monoxide, nitrogen oxide, particulate matter, and others are the primary source of pollution in metropolitan areas. These pollutants have a significant negative effect on human health and can lead to a number of conditions, including respiratory difficulties, immune system problems, and cardiovascular abnormalities (*Zhang & Batterman, 2010*). Traffic and industrial activity were reduced during the pandemic COVID-19 shutdown, which also resulted in a drop in the concentration of air pollutants. Pollutant concentrations including NO2, and particulate matter have decreased, according to satellite and ground assessments (PM). The cessation of industrial and vehicular activity was the cause of the decrease in pollutant concentration (*Bauwens et al., 2020*). The atmospheric composition and air pollution are strongly correlated with climatic variables such as humidity, temperature, and the emission of certain industrial chemicals (*Le et al., 2020*). A program known as the "sulfur oxide emission control program" reportedly began in Los Angeles in 1978. The ultimate goal was to reduce air pollution, particularly sulfur oxide emissions. There was a considerable reduction in the emission of several pollutants, including sulfur oxides, black carbon, and PM2.5, in the very following year (*Gagliardi & Andenna, 2020*).

When we compare toxic pollutants to the rise in COVID-19 infection rates, we discover that both are exactly related to one another: the greater the pollution concentration, the higher the infection rate (*Zhu et al., 2020*). Toxic pollutants in the atmosphere can induce lung disorders. A study found that during the COVID-19 shutdown, the air quality of 44 Chinese cities improved from 5 to 24 percent, with megacities like Sao Paulo witnessing a 70 percent decrease in pollution (*Krecl et al., 2020*). According to the report, industrial and traffic-related activities are the biggest contributors to air pollution. Road traffic and industrial activity were greatly diminished as a result of the shutdown. The air quality is improved as a result. Researchers have provided a case study on how the COVID-19 partial shutdown has reduced air pollution and enhanced air quality in Brazil in *Dantas et al. (2020)*. Additionally, they have looked at whether the levels of CO2, PM-10, and nitrogen

oxide decreased gradually. However, the partial lockdown has not produced sufficient results in terms of the improvement of the air quality index, thus credit should once again be given to the limited activities associated with traffic and industry.

The report on the air-quality index during the COVID-19 partial shutdown for the "Yangtze River Delta" region in northern China was provided by the researchers in *Li et al. (2020)*. The author claims that there hasn't been any improvement in the ozone layer documented, however there was evidence that indicated pollution levels were still high during lockdowns and that the causes were primarily chemical industry. Due to some factories in the northern portion of China still operating, a partial lockdown was enforced on parts of the regions, which is why the air quality index was unsatisfactory in some particular areas. The researcher did a study in China (*Wang et al., 2020*) while the country was under lockdown. Drop in transportation activity did not result in a decrease in air pollution; instead, climatic phenomena like temperature changes and moisture patterns had a major role. In addition, the author asserted that the decrease in heat waves, rain, thunderstorms, wind, lack of dust storms, tropical storms, derecho (straight-line wind), and anticyclone were the true causes of the increase in air quality. A link between the COVID-19 infection rate and variations in the weather has been established, according to a study done by *Tosepu et al. (2020)* shows due to various meteorological factors, including temperature, humidity, rainfall, air density, air pressure, and precipitation, the infection rate fluctuates throughout Indonesia.

In *Ma et al. (2020)* and *Cao et al. (2022)*, scientists investigated the relationship between the COVID-19 mortality rate and changes in the climatic conditions. In the beginning, they focused on Wuhan, China, and gathered the daily mortality records brought on by COVID-19 infection. The researchers then tracked the fluctuations in air contaminants as well as the change in climatic conditions. Based on statistical study, the correlation was created. The research investigation came to the conclusion that the increased death rate among COVID patients was also caused by the high intensity of humidity and temperature changes. Deep neural networks are regarded as being the most reliable model for the forecasting of air quality and the concentration of certain atmospheric contaminants when compared to the conventional statistical model. While performing forecasting and prediction-related activities, it is crucial to take into account the adaption of suitable models that produce accurate prediction results, even if deep neural networks are expensive in terms of computation, optimal-convergence, inefficient prediction due to noisy data, and issues related to over- and under-fitting (*Wang, Xu & Lu, 2003*).

A pruned-neural-network model that forecasts the concentration of PM10 pollution was put up by researchers in *Corani (2005)*. They conducted future projections based on past predictions using records that were largely from prior years and made up of historical data collected on an hourly basis. The aggregated-LSTM model was employed by researchers in *Chang et al. (2020)* to forecast air quality. The outcomes of LSTM were then contrasted with those of support vector regression and regression-based gradient boosting trees. LSTM outperformed other models in terms of prediction performance. *Karimian et al. (2016)* describes how researchers employed several machine-learning models to anticipate air quality by examining PM2.5 concentrations at various sizes. Multiple regression trees, deep

neural networks, and hybrid models based on LSTM were the models that were employed. However, LSTM has achieved good outcomes when it comes to forecasting and examining the efficiency results for air quality.

Various experiments have been conducted to determine how COVID-19 lock downs affected air quality. The study demonstrated how lockdown affected the air quality in six Chinese megacities. The four-month lockout had a tremendously favorable effect on the air quality. Reduced industrial activity and decreased traffic were both factors in the improvement of the air quality, which led to a 37 percent increase in the air quality index as a whole. According to a study that was published in *Shi & Brasseur (2020)*, when there is a lockdown, the concentration of ozone increases noticeably while the concentration of other environmental pollutants decreases. An Indian study revealed by *Mahato, Pal & Ghosh (2020)*, shows the air quality has improved due to the COVID-19 lock-down. The concentration of PM10 was reduced by 60%, that of PM2.5 by 39%, and the air quality as a whole was improved by 40–50%. According to a study conducted by *Menut et al. (2020)* at 126 different metropolitan locations in the UK, the lock-down time saw an improvement in the overall quality of the air, a reduction in NO2 concentration of around 48 percent, and an increase in ozone (O3) of about 11 percent.

## MATERIALS & METHODS

The LSTM model is regarded as a robust approach to handle multivariate issues (*Aljohani et al., 2021*). Using historical data, the LSTM model compares the anticipated true values of air contaminants following lockdowns.

### Data description

Data on atmospheric pollution for Lahore, Pakistan, was gathered from IQAir (https://www.iqair.com/us/pakistan/punjab/lahore). A firm in Switzerland that specializes in preserving and monitoring airborne contaminants is called IQAir. It also deals with establishing and overseeing air cleansing and quality. IQAir has received recognition for its air purifiers (*Asarch & Glendon, 2022*; *PTPA Media, 2020b*; *International Housewares Association, 2019*; *Morgan, 2019*; *PTPA Media, 2020a*) as well as for its finest air quality monitor (*BK-E Magazine, 2022*). Data from the biggest network of ground-based sensors is gathered by IQAir. To guarantee data accuracy and dependability, the organization has deployed automated and AI-based tools and sensors (air-quality meter, IOT-based air-quality monitoring system, MQ135 sensors, etc.). Phase8-DHA, Zeenat Block, Dawn Food Head Office, and HAC Agri Limited are the four stations from which data is gathered.

The hourly concentrations of various pollutants, such as PM2.5, PM10 (particulate matter), NO2 (nitrogen dioxide), NH3 (ammonia), and O3 are used to calculate the air quality data (ozone). Along with this, a few additional factors were gathered, including air temperature, air pressure, humidity, wind speed, and wind direction.

### Data pre-processing

An hourly data on a variety of air contaminants was gathered from January 2018 to May 2020. The missing values were substituted by the mean of that column for data variables

**Table 1 Station description for each atmospheric pollutant with the number of missing values.**

| | Weather Stations in Lahore City | | | |
| --- | --- | --- | --- | --- |
| Pollutants | HAC Agri Limited | Dawn Food Head Office | Phase8-DHA | Zeenat block |
| PM2.5 | 13 | 33 | 15 | 34 |
| PM10 | 22 | 15 | 56 | 74 |
| NO2 | 54 | 34 | 27 | 51 |
| O3 | 87 | 98 | 45 | 18 |

with missing values of 1,000 or less. January 2018 to May 2020 is covered by the total processed data. Table 1 lists station details as well as a list of the atmospheric contaminants and the number of missing values for each one. For the purpose of identifying the outlier and imputing its values to the column mean, we employed scatter plots and the Mahalanobis Distance (MD) to discover the outlier.

## Machine learning techniques

In machine learning, we work with a variety of algorithms that enable us to discover the link between data to produce the final prediction; the type of prediction model where we need to find prediction output in the form of continuous numerical values is known as a regression issue. Modern machine learning techniques, such as CNN, N-BEATS, decision tree regressor, support vector regressor, random forest regressor, auto-regressive integrated moving-average (ARIMA), and LSTM, have been used to explore and analyze the changes in atmospheric pollutants that occurred during the COVID-19 lockdown.

### Decision tree regressor

The main idea behind a regressor tree is to discover the point in the independent variable and divide the dataset into two parts so that the mean square error is minimized at that point. The algorithm then repeatedly accomplishes this and produces a tree-like structure.

### Support vector regressor

A supervised learning approach called the support vector regressor (SVR) is used to forecast discrete values. The hyper-plane that maximizes the number of points is the best-fit line, as determined by SVR. similar to other regression methods that aim to reduce the discrepancies between expected and actual data. Within the threshold values, SVR locates the line with the greatest match. The distance between the boundary line and the hyper-plane is known as the threshold value. SVR's fit time-complexity (number of samples) is slightly over the quadratic, which makes it challenging to scale with the dataset (having more than 1 thousand samples).

### Random forest regressor

Regression and classification issues are solved using decision trees. In the case of regression, the data flow is modelled as a tree; it begins at the root node and proceeds through splits based on the outcome variable until it reaches the leaf node, where the findings are shown. Decision trees are further divided into ensemble and bootstrapping. Bootstrapping is a

random-forest approach that builds numerous random decision trees from the data by combining ensemble learning with the decision tree. We have an output that can be used for strong classification or regression after averaging.

### Convolution neural network (CNN)

The fact that CNN supports dilated convolution, in which the filters are utilized to calculate the dilations between the cells, allows it to handle time series problems and perform tasks linked to time-series forecasting. The neural network can comprehend the association between observations of time-series data due to the magnitude of the space between the cells. The CNN architecture is capable of integrating feature-engineering into a single framework, feature extraction, and time-series representation of the data. CNN is regarded as being able to extract deep characteristics that are independent of time and being very noise resistant.

### N-BEATS

A deep stack of ensemble feed-forward networks connecting forecast and back-cast connections is the basis of the deep learning model N-BEATS. The accurate successive block model takes into account the mistake resulting from the reconstruction of the back-cast from the previous block and changes projections accordingly. Similar to the Box-Jenkins approach, this procedure fits the ARIMA model. The inner-learning and outer-learning operations make up the N-BEATS system. The inner-procedure works within the blocks and aids in local-temporal value capture in the model. The outer-learning process takes place within the stacks and aids the model in picking up on global characteristics.

### ARIMA

The ARIMA model uses prior values and its own lag predicting mistakes to describe a particular time series. Three terms, denoted by p, d, and q, make up the ARIMA model.

The autoregressive (AR) term's order is p.

The moving average (MA) term's order is q.

The amount of differencing (d) necessary to make a time series stationary is d.

The term "AR" refers to a linear-regression model that calculates its lag values; this model performs best when the predictors are unrelated to one another and do not correlate. If time-series are already stationary, then $d = 0$, which is the minimal difference necessary for stationary series. The number of delays that were employed to forecast Y is indicated by the auto-regressive (AR) term p. And q is the rank of the moving average (MA), which indicates how many forecast mistakes are included in the ARIMA model.

### LSTM

The kind of recurrent neural network (RNN) known as long short-term memory (LSTM) is capable of learning order dependency in sequencing prediction tasks. One input layer, one output layer, and a small number of completely linked hidden layers with gate units and matching memory cells make up the LSTM. Each memory cell's internal architecture ensures that the flow of errors inside its constant error-carrousel is constant (CEC). For the purpose of understanding the error-flow inside each memory cell's CEC, two unit-gates

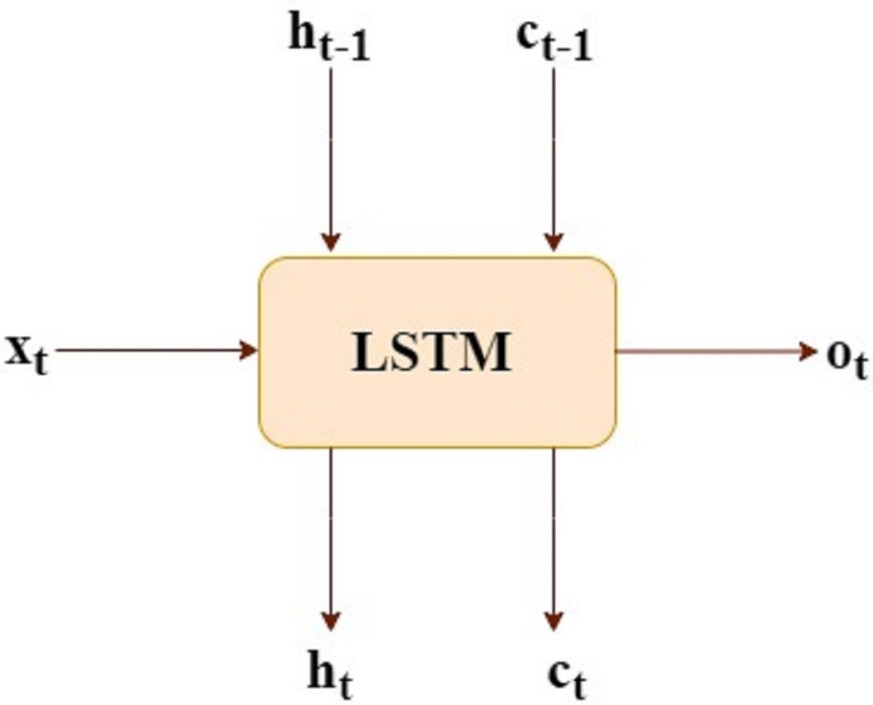

**Figure 1   LSTM mathematical form.**

open and close. The input multiplicative gate guards against irrelevant data for the CEC. The output multiplicative gate guards against irrelevant memory content for other gates.

## Mathematical model

Let LSTM has input x(t) and the output o(t)

h(t-1) and c(t-1) these are inputs from previous time step

c(t) and h(t) are consumption of next time-step, as shown in Fig. 1.

$$f_t = \sigma_g(W_f \times x_t + U_f \times h_{t-1} + b_f) \tag{1}$$

$$i_t = \sigma_g(W_i \times x_t + U_i \times h_{t-1} + b_i) \tag{2}$$

$$o_t = \sigma_g(W_o \times x_t + U_o \times h_{t-1} + b_o) \tag{3}$$

$$\hat{c_t} = \sigma_c(W_c \times x_t + U_c \times h_{t-1} + b_c) \tag{4}$$

$$c_t = f_t . c_{t-1} + i_t . (\hat{c_t}) \tag{5}$$

$$h_t = o_t . \sigma_c(c_t) \tag{6}$$

Where

$f_t$ is the forget gate

$i_t$ is the input gate

$o_t$ is the output gate

$c_t$ is the cell state

$h_t$ is the hidden state

$\sigma_g$ is the sigmoid

$\sigma_c$ is the tanh

. is the element wise multiplication

LSTM equations also generated f(t), i(t) and c ˆthese are the internal consumption of LSTM and being used for generating c(t) and h(t).

The above mathematical model shows the identical steps for one time-step, meaning all the equations must be recomputed for next time-step. Suppose we have 10 time-steps, we need to recompute the above equation for ten times. (Wf, Wi, Wo, Wc, Uf, Ui, Uo, Uc) all these are weight matrices and (bf, bi, bo, bc) are the biases and these are not time dependent.

## Exploratory analysis of data

The LSTM model was trained using data from January 2018 to December 2019. But a different collection of special data from January to February 2020, known as the validation set, was utilized for validation (VS). The lock-down set was the data from March 2020 to May 2020. (LD). There are 3,647 instances in the validation and lock-down collection, which represent the hourly AQI values of various air contaminants. To estimate the values for VS and LD, the already trained model was employed.

In order to forecast the concentration of particular contaminants, the LSTM model was used. Target and predictive variables serve as the foundation for machine learning and other deep learning approaches (*Šimić et al., 2020*). The concentration of certain air pollutants such PM2.5, PM10 (particulate matter), NO2 (nitrogen dioxide), and O3 were the goal variables (y) (ozone). These predictive factors helped the LSTM model capture the seasonal trend of traffic flow and industrial operations. The predictive variables (X) were data linked to environmental conditions coupled with their temporal variables. The RMSE, MAE, and R-SQUARE values were used to assess the model's performance.

The classification of datasets into training, validation, and lockdown time frames is shown in Fig. 2 along with dataset details. The LSTM model was trained using time series data from January 2018 to December 2019, the validation set was used for cross-validation from January 2020 to February 28, 2020, and the actual values during the lockdown period (March–May) were compared with the model predictions using least square errors. The entire research methodology is shown in Fig. 3. First, the dataset is preprocessed, then thoroughly refined, and preprocessed data is passed to the most cutting-edge machine learning models. Next, each of the machine learning models is evaluated based on RMSE, MAE, and R-SQUARE score, and finally the final results are compared. The model with the lowest error score is considered to be the best model.

## Inferential statistical analysis

To infer attributes from the dataset, inferential statistical analysis is employed. These characteristics aid in predicting future values. Different hypotheses can be tested; inferential statistics heavily relies on the testing of hypotheses. It enables us to judge if the pattern we notice is the result of chance or whether it actually has statistical significance. In order

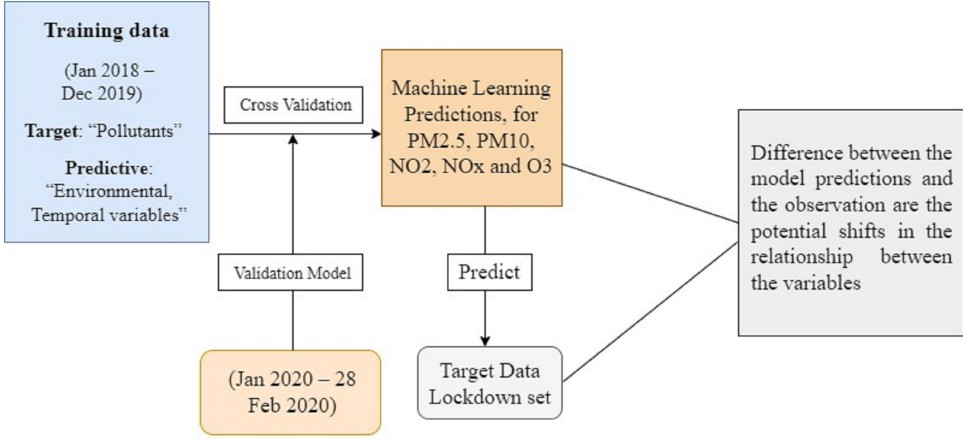

**Figure 2** **Dataset information and categorization of dataset.**

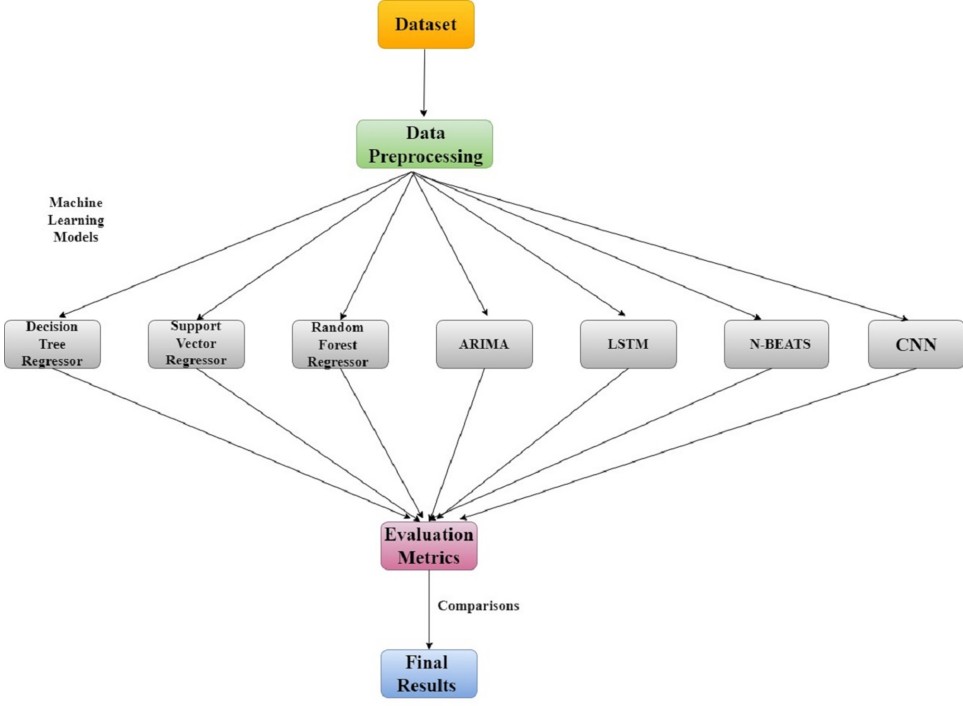

**Figure 3** **Research methodology for air quality analysis using machine learning techniques.**

to achieve this, we must first create a hypothesis and a null hypothesis. The hypothesis is the trend we expect to see in the data, while the null hypothesis is the polar opposite. In our investigation, we focused on four air pollutants: PM2.5, PM10, NO2, and O3. We will analyze how these contaminants impact the air quality index (AQI). According to our theory, the concentration of these contaminants and the AQI are linearly correlated. The null hypothesis demonstrates that there is no linear relationship between these pollutants'

**Table 2  Inferential statistical analysis for different pollutants.**

| Pollutants | R-squared | Adj. R-squared | F-statistic | slope | intercept | P > \|t\| |
|---|---|---|---|---|---|---|
| Pm2.5 | 0.089 | 0.085 | 1.08 | −0.0014 | 0.322 | 0.000 |
| Pm10 | 0.270 | 0.267 | 3.27 | −7.6301 | 0.269 | 0.000 |
| No2 | 0.006 | 0.002 | 0.234 | −0.008 | 0.2316 | 0.000 |
| O3 | 0.052 | 0.048 | 0.00023 | −0.0014 | 0.3056 | 0.000 |

concentrations and AQI. Following that, we do various tests to determine if there is statistical significance. Table 2 demonstrates that the *p*-value for our parameters is less than 0.05 and that the F-statistics for PM2.5 is substantial. The null hypothesis may thus be rejected as we can presume that there is statistical significance. Similar circumstances apply to the pollutants PM10, NO2, and O3. In Table 2, the negative slope indicates that the air quality index (AQI) will decline as pollutant concentration rises and increase as concentration falls.

### Validation method and comparison

The projected concentration of each air contaminant is shown by the LSTM findings. These predicted concentrations and the contaminants' actual values were contrasted. Both from a long-term and short-term viewpoint, the evolution of the historical data was compared to our technique. Due to a decrease in traffic and industrial activity, we see a decrease in the concentration of air pollutants.

## EXPERIMENT AND RESULTS

The key concern is using deep learning to research how COVID-19 lockdown affects air quality. The algorithms use historical data from January 2018 to May 2020 to forecast the air quality during the shutdown. The historical records were further divided into two categories: lockdown set, and validation set. The historical records prior to March 2020 were used for validation, and the data from March through May 2020 displays the lockdown set that is used for testing. The data from January 2018 through December 2019 was used for training the models. Models are employed to compare each pollutant's projected levels to its actual values. The RMSE, MAE, and R-SQUARE values are used to assess models. In comparison to other models, the LSTM model demonstrated the fewest errors and the best predictions for the lockdown and validation time frames.

### Explorative analysis

We use explorative data analysis to determine the relative importance of each air contaminant in order to comprehend the relationships between them. We employ principal component analysis for that (PCA). For the best outcomes, it is advised to select the components that account for between 70 and 80 percent of the overall variation based on explained variance. We select five elements from the data that are the most beneficial and also include important information by keeping the concept in mind.

## Hyper-parameter tuning for LSTM

To influence the behavior of deep learning and machine learning models, hyper-parameter tweaking is crucial. Our model's ability to produce accurate results depends on our ability to fine-tune its hyper-parameters. If we are unable to minimize the loss-function, our model will produce more mistakes. We have adjusted the hyper-parameters for the LSTM model while keeping this idea in mind. The LSTM model has four layers: one input layer, two hidden layers, and one output layer. It is generally recommended to utilize two hidden layers for complicated problems, and we have done the same. Second, we concentrate on the number of nodes. When we lower the number of nodes, it improves accuracy and, in other words, reduces overfitting. Removing nodes act as a regularizer. A total of 50 nodes were employed in the input layer, while 50% fewer nodes were used in the second and third layers. We used the ReLU activation function on the hidden layers because ReLU is nonlinear and has the advantage of not having back propagation error, and we used the tanh activation function on the output layer because the output from that layer ranges from $-1$ to 1, and tanh is thought to be a good one for solving regression problems. In our situation, after 100 epochs the validation accuracy started to fall, therefore, we set epochs to 100. As Adam optimizer incorporates the best qualities of RMSProp and AdaGrad algorithms, it can handle the sparse gradients on noisy data extremely quickly. The learning rate was set to 0.1. We utilized root mean square error and the Adam optimizer as measurements. Although we have used LSTM with a variety of settings, the results are greatest when using the fine-tuned parameters stated above.

## Evaluation parameters

We employed several machine learning assessment criteria. RMSE, MAE, and R-SQUARE score were among the metrics used.

### *Root mean square error*

The standard-deviation of the prediction errors is known as RMSE (known as residuals). The prediction errors show how far apart from the regression line the data points are. The concentration of data around the best-fit line is measured by RMSE.

$$RMSEf_o = \sqrt[2]{[\sum_{i=1}^{N}(Z_f i - Z_o i)^2 / N]} \tag{7}$$

Where: $\sum$ = summation ("add up")
  $(Z fi–Zoi)^2$ = differences, squared
  N = Normal size

### *Mean absolute error*

Our second evaluation parameter is mean absolute error, and absolute error is defined as the magnitude of difference between the mean value and each individual value. It can also be defined as the arithmetic mean of all the absolute errors is taken as the mean absolute error of the value of that quantity.

$$MAE = 1/n \sum_{i=1}^{n} |x_i - x| \tag{8}$$

Where:

$|x_i - x|$ = absolute-error

n = number of errors

### R-SQUARE score

The third parameter is the R-SQUARE score, defined as the statistical measures that represent the proportion of the variance for a dependent variable that is explained by an independent variable or a variable in a regression model. In contrast, the correlation shows a strong relationship between the dependent and independent variables. It also shows the extent to which a variable explains the variance of another variable.

$$R^2 = SS_{regression}/SS_{total} \tag{9}$$

Where

$SS_{regression}$ is the sum squares by the regression

$SS_{total}$ is the total sum of squares

### Deep learning results

We have trained various machine learning models, including decision tree, SVR, random forest, ARMIA, CNN, N-BEATS, and LSTM, on the entire training dataset, which contains historical records from January 2018 to December 2019, in order to better understand the true changes in the concentration of various atmospheric pollutants. Additionally, these models were fitted independently to the data from January 2020 to May 2020 (validation set and lockdown set). The results were then assessed using the mean absolute error (MAE), R-SQUARE Score, and root mean square error (RMSE). When compared to other models, the LSTM model performs best and has the fewest mistakes. The primary explanation for this is because LSTM excels in time-series forecasting and develops forecasts based on prior sequential data. Compared to other models, LSTM has a remarkable capacity to efficiently remember data.

Figure 4 displays the LSTM model's training and validation losses in terms of mean square error and root mean square error. On one hundred epochs, the initial training was run. After 10 epochs, the training and validation losses are consistent. During the first stages of training, the error was at its highest point and the gap between the two losses was at its greatest.

### Comparison of LSTM model with other machine learning models

In relation to various pollutants, the overall findings of several machine learning models are shown in Table 3. On the air quality dataset, we applied a variety of machine learning models, including decision tree, SVR, random forest, ARIMA, CNN, N-BEATS, and LSTM. Error metrics including RMSE, MAE, and R-SQUARE score were used to evaluate the machine learning models. The top four pollutants accounted for 70% of the overall variance when we used PCA to choose features. The contaminants that were chosen were PM2.5, PM10, NO2, and O3. For each pollutant, we have utilized a scatter plot and the Mahalanobis Distance (MD) to identify the outlier and impute its values using the meaning of the column. For each pollutant, we modified the LSTM model's parameters and put

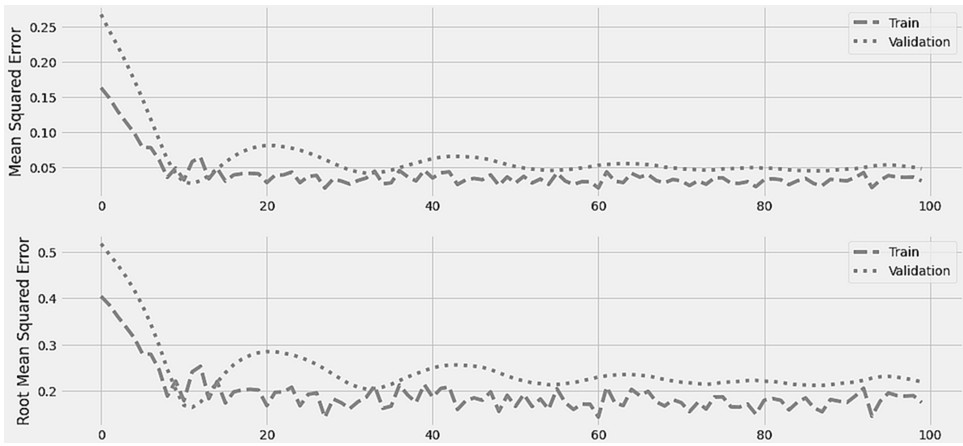

**Figure 4 LSTM training and error minimization against 100 epochs.**

other cutting-edge machine learning models into practice. The findings were then created based on error measurements. All of the machine learning models worked well, however the LSTM model excelled in comparison to the others. The LSTM model has the lowest error for each of the various pollutants when compared to other models. Since we are projecting the pollutant concentration and utilizing time series data, the LSTM network is the ideal model for time series data. LSTM has the capacity to predict future values based on prior sequential data (past historical records). When compared to other machine learning models, LSTM performed better since it can anticipate the future while learning the prior levels of various contaminants. PM2.5 pollutant had a score of 4.46 percent MAE, 0.25 percent R-SQUARE score, and 9.99 percent RMSE from the LSTM model. The PM10 pollutant's LSTM score was 12.84 percent RMSE, 0.75 percent R-SQUARE score, and 8.58 percent MAE. The results of the LSTM model for NO2 were 4.35 MAE, 0.12 R-SQUARE, and 6.77 RMSE. 8.20 percent MAE, 0.17 percent R-SQUARE, and 29.79 percent RMSE made up the O3 LSTM score.

The four parts that make up an elementary LSTM unit are as follows: a cell, an input gate, an output gate, and a forget gate. A cell's function is to retain values for any given period of time. Additionally, gates' function is to control how information enters and exits a cell. For time-series forecasting, LSTM consistently delivered good results. Sequential data's future values are effectively predicted. There is no need for precise modification because the LSTM gives us a wealth of parameters, including input, output, biases, and learning rates. The LSTM model outperforms ARMIA in terms of performance. The major causes of this are because ARMIA is computationally costly and performs poorly for situations involving seasonal time-series and long-term forecasting. While LSTM often works with large datasets, where sufficient data must be provided for initial model training, ARIMA excels on smaller datasets, but it might be challenging to pinpoint the ARIMA model's pivotal moments. When it comes to decision trees and random forests, a forest needs a lot of memory since it stores data from hundreds of trees. The LSTM cell stores information, and gates control that memory. Similarly, visualize the series of choices that results in

**Table 3** Machine learning results for different pollutants.

| Models | Pollutants | MAE | R2 | RMSE |
|---|---|---|---|---|
| Decision Tree | PM2.5 | 5.327 | 0.283 | 94.607 |
| | PM10 | 35.403 | 0.434 | 42.186 |
| | NO2 | 6.824 | 0.582 | 154.417 |
| | O3 | 17.464 | 0.143 | 716.448 |
| SVM | PM2.5 | 10.752 | 1.375 | 175.17 |
| | PM10 | 37.815 | 0.961 | 326.837 |
| | NO2 | 5.537 | 0.482 | 102.296 |
| | O3 | 26.611 | 1.464 | 118.278 |
| Random Forest | PM2.5 | 8.352 | 1.082 | 153.574 |
| | PM10 | 38.447 | 0.125 | 332.532 |
| | NO2 | 6.076 | 0.105 | 107.886 |
| | O3 | 17.714 | 0.194 | 57.125 |
| N-BEATS | PM2.5 | 16.47 | 2.69 | 21.42 |
| | PM10 | 18.25 | 4.61 | 19.78 |
| | NO2 | 7.12 | 4.22 | 8.47 |
| | O3 | 11.41 | 3.37 | 31.58 |
| ARIMA | PM2.5 | 11.926 | 0.961 | 6.41 |
| | PM10 | 9.241 | 0.76 | 41.809 |
| | NO2 | 7.937 | 0.2294 | 32.023 |
| | O3 | 4.143 | 0.895 | 36.979 |
| CNN | PM2.5 | 16.54 | 1.25 | 7.57 |
| | PM10 | 11.26 | 1.04 | 14.92 |
| | NO2 | 8.58 | 0.66 | 11.24 |
| | O3 | 9.16 | 0.74 | 39.41 |
| LSTM | PM2.5 | 4.46 | 0.25 | 9.99 |
| | PM10 | 8.58 | 0.75 | 12.84 |
| | NO2 | 4.35 | 0.12 | 6.77 |
| | O3 | 8.2 | 0.17 | 29.79 |

computational burden. That is a reliable answer. The SVM method works well for small datasets, but it struggles with noisy data, data where target classes overlap, or data where the number of training data samples is insufficient for the number of features. As a result, these algorithms do not perform as well as LSTM for air quality datasets.

## LSTM results with respect to different pollutants

Figure 5 shows the PM2.5 concentration for the period from January to May. The predicted values, which were compared to the actual values. The time series data from January to May are displayed on the plot. The validation set is from January 2020 to February 28th, and the lockdown period is from March 2020 to May 2020. The graph's analysis reveals that there is no average decrease in PM2.5 concentration from January to the end of February.

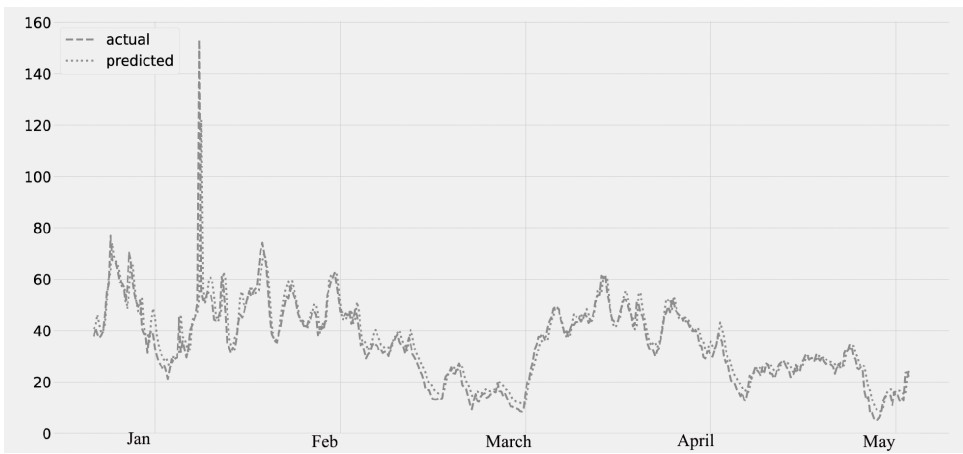

**Figure 5** Time series plot for the concentration of PM2.5 pollutant (Jan 2020–May 2020).

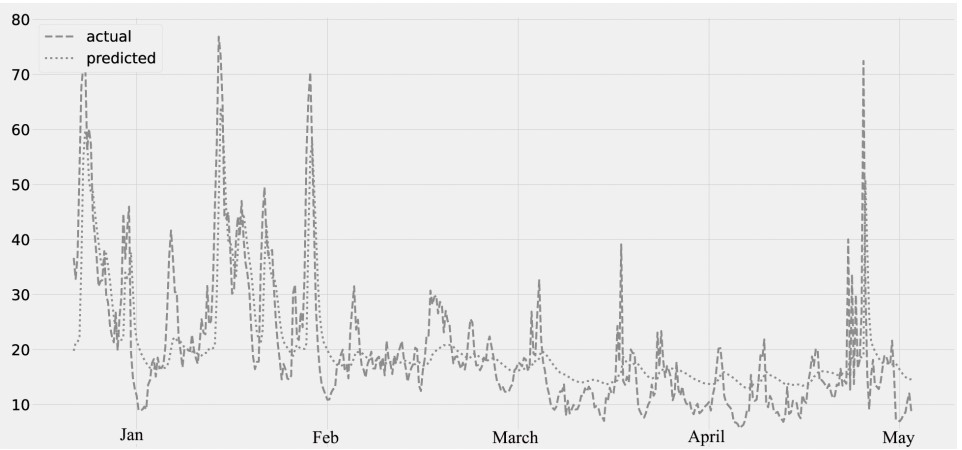

**Figure 6** Time series plot for the concentration of NO2 pollutant (Jan 2020–May 2020).

However, there is a decrease in the concentration of PM2.5 after the middle of March (the lockdown timeframe).

The concentration of NO2 for the period from January to May is shown in Fig. 6. The predicted values, assessed against the actual values. The validation set is from January through February 28th, and the lockdown period is from March through May of the following year. The concentration of NO2 decreases from the middle of March to the end of May. The graph reveals a steady decline in air pollution after March, which appears to be the true result of the COVID-19 lockout.

Figure 7 displays the PM10 concentration. The predicted values, which were compared to the actual values. The figure displays the time series data as well as the PM10 concentration. The validation set is the period from January 2020 to February 2020; the lockdown period

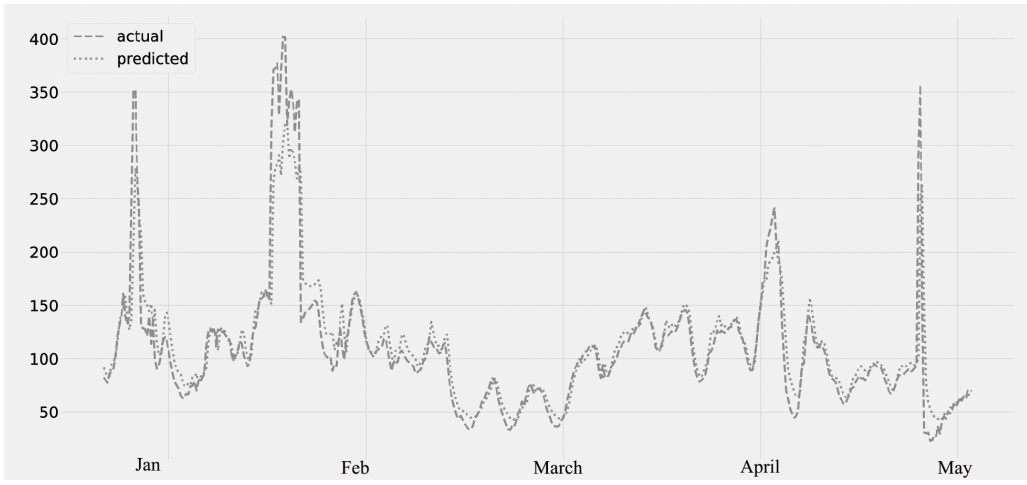

**Figure 7   Time series plot for the concentration of PM10 pollutant (Jan 2020 –May 2020).**

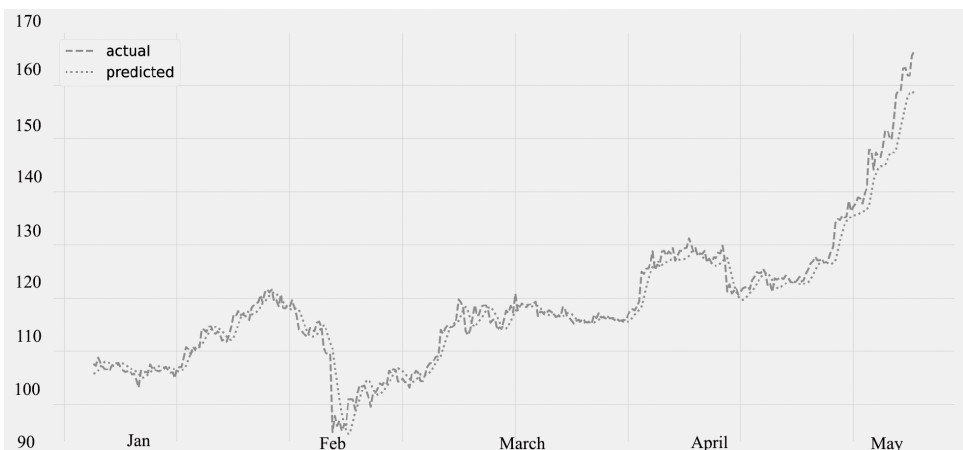

**Figure 8   Time series plot for the concentration of O3 (Jan 2020–May 2020).**

is from March 2020 to May 2020. It was noticed that the amount of PM10 pollutants was falling.

Figure 8 displays the ozone (O3) levels for various time periods. The predicted values, which were compared to the actual values. The statistics for several time periods from January through May are represented on the figure. The validation set is from January through February 28th, and the lockdown period is from March through May of the following year. The graphic unequivocally demonstrates that the amount of ozone is rising (O3). The quality of ozone has significantly improved as a result of the concentration of some air contaminants being lower. There is a decrease in the concentration of air contaminants, as seen in Figs. 3, 6 and 7.

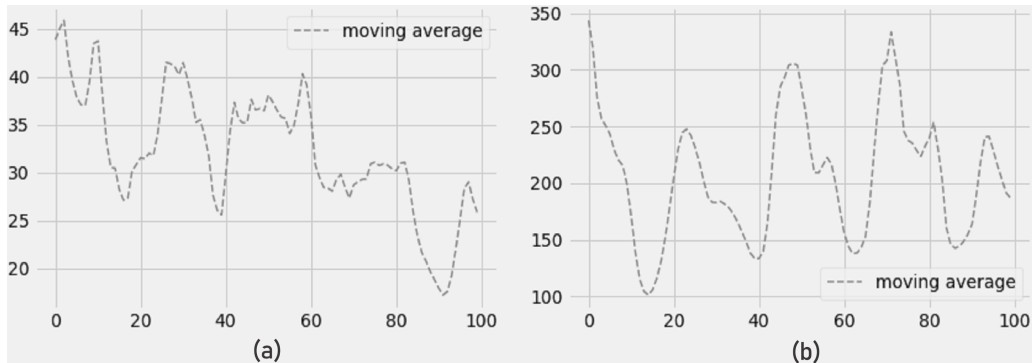

**Figure 9** (A) Moving average PM2.5, and (B) moving average PM10.

## Forecasting moving average

The first stage in the forecasting process is to ascertain the maximum number of hours the model can anticipate accurately. Most of the time, the extended lead-time may cause performance that is ineffective and inaccurate. The R-SQUARE error levels determine how accurate the results are. R-SQUARE error levels increase as predicting time increases. On the other side, a shorter predicting period results in the least number of mistakes. In light of this, we have predicted the concentration of several pollutants, including PM2.5, PM10, NO2, and O3, over the next four days using a moving average.

The 4-day moving average of PM2.5 and PM10 pollutants is shown in Figs. 9A and 9B, and the findings show a reduction in PM2.5 and PM10 pollutant concentrations for the next four days. The error measurements have not been impacted by the moving average. The LSTM model still shows minimum errors in terms of RMSE, MAE, and R-SQUARE score while forecasting 4 days of moving average.

Figure 10A depicts a drop in NO2 concentration and Fig. 10B depicts an increase in O3 concentration (ozone). In terms of RMSE, MAE, and R-SQUARE score, the 4-day prediction of moving average produces strong results with little mistakes.

## Reduction in air pollution

Comparing the lockdown of 2020 to the same time period in the preceding years, Table 4 provides a clearer picture of certain contaminants and their average concentrations. We calculated the average concentration for each pollutant and displayed it in Table 4. For our research investigation, the well-known contaminants that were thought to be the more serious causes of pollution were selected. The contaminants that were considered were PM10, PM2.5, NO2, NOx, and O3. These contaminants' typical concentrations were measured and contrasted with values from the same period in the prior years. Table 4 displays time series data for three months and compares the average concentration of various pollutants during the lockdown periods (March 2020 to May 2020) to the corresponding periods for the two years prior, 2018 and 2019, as well as the average concentration of various pollutants chosen for the years of 2018, 2019, and 2020. (March,
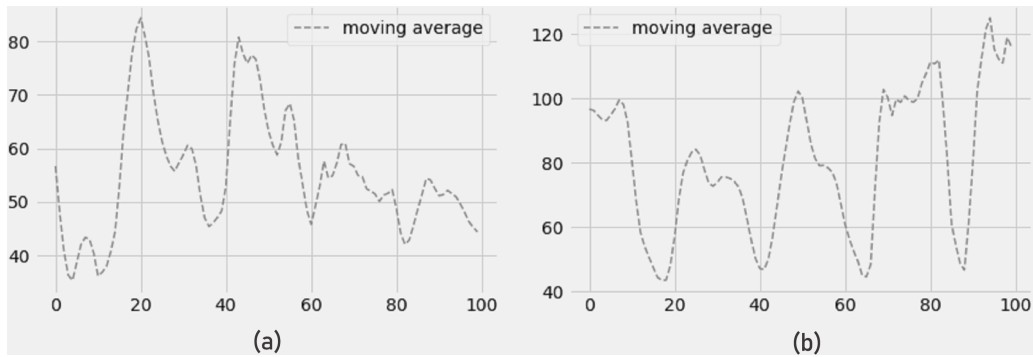

**Figure 10** Moving average of (a) NO2 concentration and (b) O3 concentration.

**Table 4** Average concentration of different pollutants during lockdown (March 2020–May 2020) and same time frame in previous years (March–May in 2018 and March–May 2019).

| Average variables | 2018 | | | 2019 | | | 2020 | | |
|---|---|---|---|---|---|---|---|---|---|
| | March | April | May | March | April | May | March | April | May |
| PM2.5 | 119.46 | 157.78 | 179.75 | 166.78 | 154.23 | 169.43 | 114.32 | 96.32 | 71.68 |
| PM10 | 144.72 | 173.72 | 184.72 | 153.65 | 175.26 | 188.82 | 141.55 | 99.46 | 86.17 |
| NO2 | 34.67 | 39.44 | 47.64 | 52.87 | 46.43 | 59.49 | 51.6 | 26.36 | 19.07 |
| NOx | 11.6 | 11.32 | 13.6 | 18.65 | 20.16 | 27.65 | 20.39 | 12.63 | 8.02 |
| O3 | 42.72 | 38.23 | 31.09 | 34.03 | 32.4 | 31.72 | 21.11 | 29.1 | 35.5 |

May, and April). We found that the concentration of PM2.5 pollutants was 31% lower in March 2020 than it was in March 2019 after examining these statistical data.

When the PM2.5 pollutant concentrations in April (2019 and 2020) were compared, April 2020 had a 33 percent lower pollutant concentration than April 2019. When May 2019 and May 2020's PM2.5 concentrations were compared, it was found that the latter month's reading was 57 percent lower. Similar to March 2019, there was a 7.8% decrease in PM10 concentration in March 2020 compared to March 2019. When compared to April 2019, there was a 42 percent decrease in April comparing March 2020 to March 2019, the concentration of NO2 declined to 1.92 percent, by 43 percent in April 2020, and by 67 percent in May 2020. In comparison to March 2019, April 2020, and May 2019, the average NOx concentration reduced by 11.1 percent, 40 percent, and 70 percent, respectively. The quality of the air will improve as the number of atmospheric pollutants decreases, improving the overall concentration of ozone O3 during the lockdown period. In comparison to March 2019, there was a 38 percent rise in ozone (O3) in March 2020. Ozone (O3) levels increased for the month of April 2020 by 9.3% compared to April 2019. In addition, ozone (O3) increased by 12 percent in May 2020 compared to May 2019 within the same calendar year. When we compared the levels of several contaminants in 2020 and

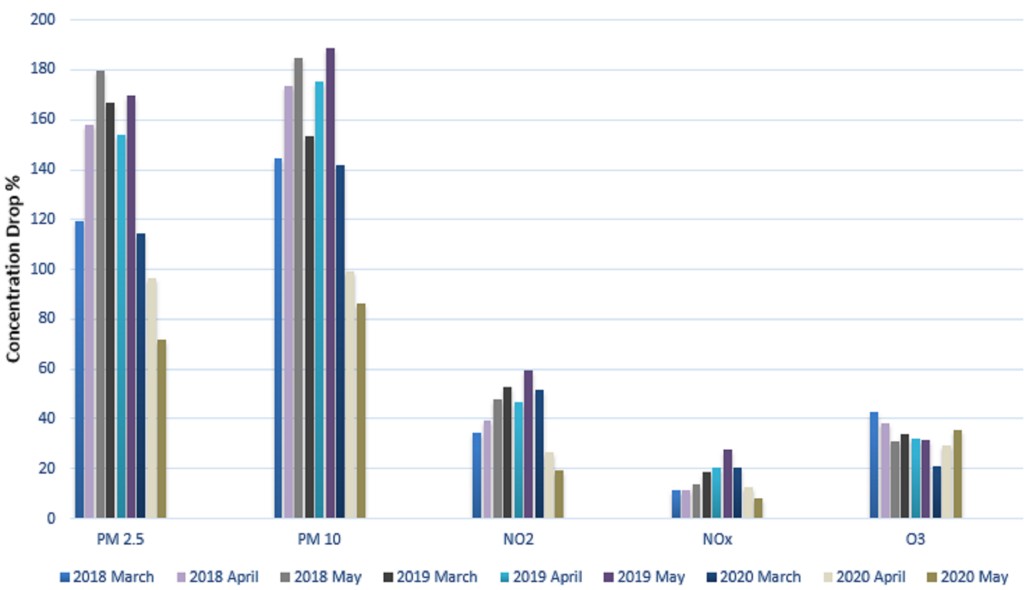

**Figure 11 Average concentration of different atmospheric pollutants.**

2018, there was a considerable drop. Figure 11 provides a clear visual illustration of Table 4's graphical form.

Figure 11 shows a visual depiction of the average concentration of several air pollutants during the months of March, April, and May for the years (2018), 2019 and 2020.

# CONCLUSION AND FUTURE WORK

In this work, we examined variations in the concentration of several air pollutants during the Lahore COVID-19 lockdown (Pakistan). The LSTM model was used in the study to analyze the discrepancy between real and forecasted values for the concentration of various contaminants. The LSTM model's prediction was more than enough for examining variations in the concentration of several contaminants. During the lockdown, the number of pollutants was significantly reduced, while the air's quality (Ozone O3) was enhanced. Nevertheless, because of decreased industrial activity and traffic during the lockdown in the Pakistani city of Lahore, the ''Air Quality Index'' as a whole improved. The findings show that the LSTM model has high potential and is regarded as a suitable tool for investigating and assessing the changes occurring to air contaminants during the COVID-19 shutdown. We came to the conclusion that deep learning is the most reliable approach for researching and monitoring time series issues. However, further research is needed to examine the changes taking place in time series issues on a larger scale. We will research deep learning models with enhanced generalization capabilities in the future to address time-series issues. Moreover, we intend to make a generic prediction of the air quality by taking the dataset of post COVID from different cities of Pakistan.

### Funding

The authors received funding from the Deanship of Scientific Research at King Khalid University through large groups (Project under grant number (235/1444)). The funders had no role in study design, data collection and analysis, decision to publish, or preparation of the manuscript.

### Grant Disclosures

The following grant information was disclosed by the authors:
Deanship of Scientific Research at King Khalid University: 235/1444.

### Competing Interests

The authors declare there are no competing interests.

### Author Contributions

- Umer Zukaib performed the experiments, performed the computation work, prepared figures and/or tables, and approved the final draft.
- Mohammed Maray analyzed the data, authored or reviewed drafts of the article, and approved the final draft.
- Saad Mustafa conceived and designed the experiments, performed the computation work, prepared figures and/or tables, and approved the final draft.
- Nuhman Ul Haq conceived and designed the experiments, performed the computation work, prepared figures and/or tables, and approved the final draft.
- Atta ur Rehman Khan analyzed the data, authored or reviewed drafts of the article, and approved the final draft.
- Faisal Rehman performed the experiments, prepared figures and/or tables, and approved the final draft.

### Data Availability

The code is available in the Supplementary File.

### Supplemental Information

Supplemental information for this article can be found online at http://dx.doi.org/10.7717/peerj-cs.1270#supplemental-information.

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
