# Peer review of "Impact of COVID-19 lockdown on air quality analyzed through machine learning techniques"

_PeerJ Computer Science, doi:10.7717/peerj-cs.1270_

## Round 0.1 · original submission · Major Revisions

We invite you to submit a majorly revised manuscript addressing each reviewer's comments. Also, address the following comments:
1. Improve the abstract and provide numerical results to support how the LSTM model outperformed.
2. In section 3.1, describe the data source further, especially its reliability. Why did you select this data source: https://www.iqair.com/us/pakistan/punjab/lahore. How do they collect and verify the data?
3. How did you select these five machine learning models? Describe each model further.
4. Define/expand each abbreviation at its first appearance/use in the manuscript.
5. Thoroughly revise the manuscript to correct all grammatical/typo/style/spelling errors.

·

Basic reporting

Corrections are needed esp. in grammar and expression:
Line 59: In this paper, we proposed a comparative study in order to
Change to we present a comparative study
Line 162 is a densely populated urbanised city in 163 Pakistan (Sentence is incomplete, the starting city name is missing)
A standard should be followed for the names of Months: Sometimes its written as Jan. sometimes as Jan
Line 187 Figure.1 1 needs to be written properly
R2 = 1− RSS / T SS in Equation 3, Variable needs to be written as R2
Line 260 needs to be rewritten as “ LSTM is always performs well for time-series forecasting”
Figure 2 needs to be labeled correctly

Experimental design

The authors need to suggest their own contribution in the experimental methods.
As its a comparative study, the implementations of existing algorithms and hyperparameters need to be discussed in more detail.

Validity of the findings

No comment

Additional comments

Experiments can be performed with different parameters for the same algorithm to decide the best match with observed results.

Reviewer 2 ·

Basic reporting

ok

Experimental design

No figure showing detail research methodology is presented. Mathematical modeling for proposed method using LSTM must be included

Validity of the findings

The result shows comparison between different models is represented , however justification of the findings need to be clearly stated.

Additional comments

The problem definition is well defined. The data description and pre-processing need to be given in more detail. Figure 1 should include all detail process of research methodology. Line no 187 is in error. Figure no is wrong. The result discussion should clearly bring out the justification of using LSTM for the current problem. Also include mathematical expressions of LSTM model.

·

Basic reporting

Authors propose an interesting approach to analyse the impact of COVID-19 lockdown on Air Quality. While the basic idea is technically sound and enjoys good novelty, a few concerns are raised, and details are as follows:
Please give detail introduction about the research motivation, which is less comprehensive. It would be great to see:
1) why the research problem studied is important, especially what could be key impacts in real world applications and theoretical exploration?
2) what are the main characteristics (con/pro) for existing approaches?
3) Authors need to discuss the comparative with Deep Learning models. Why the are stick with the conventional approaches.

Experimental design

4) The use of SVM, decision tree and random forest in case of regression problem is questionable. The problem belongs to regression problem and the comparative methods are the solution who perform well in classification problems.
5) Why is LSTM selected as base classifier in the proposed model?
6) There could have been some inferential statistical analysis to be performed on the parameters of Air quality index. This will also explore the latent relation exist amongst the attributes.
7) The performance of the proposed approach is better when compared to the conventional model but what about the state-of-the-art model prediction comparison.
8) Why not the other environmental factors are not considered for the analysis of Air quality such as Traffic, weather, location etc.

Validity of the findings

9) what is the measure taken to handle outliers since the event like lockdown will not be a frequent event, it will happen very rarely so measuring the air quality across the year and also for the lockdown situation there must be appropriate measures to be taken.
The work is also close to the aims and scope of the journal and so, I think it can be accepted for publication in its revised version. Prior to publication it is important to further improve the use of English language in the manuscript.

---

## Round 0.2 · Minor Revisions

I appreciate the authors for revising the manuscript. I invite the authors to revise the manuscript further, addressing the reviewer's comments.

·

Basic reporting

The equations need to be numbered.
Also check the R2 equation again.( Parameter is written as R square)

Experimental design

No Comment

Validity of the findings

No Comment

Additional comments

The paper is updated as per the inputs.
The graph shown in Figure 4 is not clear as it has two graphs with similar pattern, though it is given in text . If the Y-axis can be defined with the variable being represented it will be better.

·

Basic reporting

Language has been substantially improved and its satisfactory now.

Experimental design

Response given by the author are satisfactory

Validity of the findings

Authors has validated the finding by performing various experiments.

---

## Round 0.3 · accepted · Accept

Congratulations on addressing all reviewers' comments. I am glad to inform you that your paper has now been accepted.

·

Basic reporting

NO Comment

Experimental design

NO Comment

Validity of the findings

No Comment

Additional comments

The paper is now updated as per the suggestions of all reviewers